# Effects of rivastigmine on gait in patients with neurodegenerative disorders: A systematic review and meta-analysis

Sung Ryul Shim[1], Jong-Yeup Kim[1,2], Kyum-Yil Kwon[3], Jieun Shin[1], Yungjin Lee[4], Seon-Min Lee[5,6]*

1 Department of Biomedical Informatics, Konyang University College of Medicine, Daejeon, Republic of Korea, 2 Department of Otorhinolaryngology-Head and Neck Surgery, Konyang University College of Medicine, Daejeon, Republic of Korea, 3 Department of Neurology, Soonchunhyang University College of Medicine, Seoul, Republic of Korea, 4 Department of Rehabilitation Medicine, Konyang University College of Medicine, Daejeon, Republic of Korea, 5 Department of Neurology, Konyang University College of Medicine, Daejeon, Republic of Korea, 6 Myunggok Medical Research Institute, Konyang University College of Medicine, Daejeon, Republic of Korea

☯ These authors contributed equally to this work.

* nestoml@kyuh.ac.kr

**Data Availability Statement:** All relevant data are within the paper and its Supporting information files. Some or all data, models, or code generated

## Abstract

### Background & aims

Gait disturbances are commonly observed in patients with neurodegenerative disorders, including Alzheimer's disease, Parkinson's disease, and higher-level gait disorders, which are associated with cholinergic deficits. We conducted a systematic review and meta-analysis to investigate the effects of rivastigmine on improvement in gait.

### Methods

A comprehensive literature search was conducted using Medical Subject Heading (MeSH) terms and text keywords related to gait and falls after rivastigmine treatment for neurodegenerative disorders. The intervention (rivastigmine), comparison (control or no treatment), and outcomes of improvement in gait speed and fall were assessed from database inception to April 2024. References and collected data were meticulously reviewed to ensure the integrity of the included studies. Standardized mean differences (SMDs) and Hedges'g, along with their 95% confidence intervals (CIs), were calculated for gait speed and number of falls.

### Results

A total of 222 articles were identified during the initial search across different electronic databases, 50 including PubMed (n = 23), Cochrane (n = 19), Embase (n = 139), Scopus (n = 38), and a manual search (n = 3). Finally, we conducted a systematic review and meta-analysis focusing on the final four studies, encompassing 286 participants. The pooled SMD for the overall gait speed without a comparison group was 0.761 (95% CI: −1.165 to 2.688), indicating no significant improvement in gait speed. For the overall

or used during the study are available from the corresponding author upon request.

**Funding:** The author(s) received no specific funding for this work.

**Competing interests:** The authors have declared that no competing interests exist.

number of falls between the rivastigmine treatment and control groups, the pooled SMD was −0.366 (95% CI: −0.650 to −0.083). A statistically significant reduction in the number of falls was observed in the rivastigmine group than in the control group.

## Conclusion

Rivastigmine treatment in patients with neurodegenerative disorders tend to improve gait speed and significantly reduces fall incidence. Given the limited efficacy of current treatments for gait disturbances and falls, dual cholinesterase inhibitors like rivastigmine could be a promising therapeutic option.

## Introduction

Gait disturbances are commonly observed in the older individuals and can arise from various causes. However, they are particularly prevalent in individuals with neurodegenerative disorders. The most devastating consequences of gait disturbances and postural instability are falls, which lead to fractures and traumatic brain injuries, thus becoming a major contributor to increased mortality rates and underscoring the urgency of treatment [1]. The maintenance of gait and postural control necessitates the intricate integration of complex sensorimotor functions, for which cognitive abilities play a pivotal role. Furthermore, a robust association has been established between the severity of cognitive impairment and the extent of gait disturbance.

Some studies have suggested a close association among impaired attention, executive function, and gait disturbances, highlighting choline deficiency as a significant factor [2]. Other observations have linked reduced cortical acetylcholinesterase (AChE) activity to falls [3]. Additionally, a recent neuroimaging study revealed that the brainstem pedunculopontine nucleus cholinergic system, along with basal forebrain cholinergic system abnormalities, play a crucial role in gait and falls [3, 4]. Therefore, cholinergic deficits contributing to gait and cognitive dysfunction in patients with neurodegenerative disorders may provide rationale for pharmacological interventions for these conditions.

Cholinesterase inhibitors (ChEIs) are widely used to improve the cognitive function in patients with dementia. Recent clinical studies have investigated the impact of ChEIs on gait, postural instability, and falls, demonstrating their potential benefits. Several studies have provided evidence supporting the notion that ChEIs improve executive function and attention, thereby reducing the frequency of falls [5–7]. Additionally, among ChEIs, rivastigmine is unique due its additional mechanism of action as a butyrylcholinesterase (BuChE) inhibitor. It has been proposed as an effective medication based on evidence derived from studies involving patients with Parkinson's disease (PD) having cognitive impairment. Consequently, several studies have anticipated the effect of rivastigmine on gait problems accompanying cognitive impairments in disorders other than PD. Based on these findings, several controlled randomized clinical trials targeting patients with cholinergic deficits have recently been reported, along with systematic reviews and meta-analyses of the effects of ChEIs on gait [8, 9]. However, there is still insufficient clarity on the effects on gait and falls. Furthermore, despite several studies, systematic reviews, and meta-analyses examining the cognitive improvements associated with rivastigmine, research on its effects on gait is lacking. Therefore, this study aims to conduct a systematic review and meta-analysis to investigate the effect of the rivastigmine on gait improvement in patients with neurodegenerative disorders (S1 Fig).

## Materials and methods

This systematic review and meta-analysis was registered in the PROSPERO database (registration number: CRD42023494662) and thoroughly complied with the Preferred Reporting Items for Systematic Reviews and Meta-Analyses (PRISMA) statement (S1 Table) [10].

### Data sources and literature search

A comprehensive literature search was conducted in the PubMed, MEDLINE, Embase, Cochrane, and Scopus databases using Medical Subject Headings (MeSH) terms and text keywords related to gait and falls after rivastigmine treatment for neurodegenerative disorders that could affect cognitive impairments and/or dementia, intervention (rivastigmine), comparison (control or no treatment), and outcomes of gait speed and fall improvement from database inception to April 2024 (S2 Table). The search terms were grouped using Boolean operators (e.g., AND, OR, and NOT). The literature search did not place any restrictions on language or study design. Two independent researchers (SRS and S-ML) manually searched for all relevant studies conducted in clinical trial registries and Google Scholar.

### Study selection

The inclusion criteria were as follows: (1) studies including patients who had neurodegenerative disorders that can affect cognitive impairments such as Alzheimer's disease (AD) and Parkinson's disease (PD), and/or higher level gait disorders, (2) intervention included prescription of rivastigmine, (3) comparisons were specified as with the inclusion of a control group for evaluating the number of fall or were not specified, focusing solely on the effects of rivastigmine treatment on gait speed, and (4) outcomes were measured as mean differences in gait speed and the number of fall. To ensure data accuracy and relevance, duplicate publications and publications that did not contain original articles (conference abstracts only, case reports, case series, review articles, editorials, letters, and guidelines) were excluded from the analysis. Two independent investigators (SRS and S-ML) primarily analyzed the titles and abstracts, and then reviewed full-text articles according to the inclusion and exclusion criteria. A data extraction form was used independently by the authors to extract data. The final inclusion of articles was determined through collaborative evaluation meetings involving all authors. In addition, to ensure the integrity of the included studies, the references and collected data were reviewed meticulously so that they did not overlap (S3 and S4 Tables).

### Data extraction

Basic details about the studies (first author, year of publication, country, study design, and number of patients), patient characteristics (age, sex, and disease), and technical aspects (treatments and controls) were extracted from the included articles using a predefined data extraction form. The final meta-analysis only included studies that provided comprehensive and complete information.

### Meta-analysis assessment of outcome findings and statistical analysis

The standardized mean difference (SMD, Hedges' g) along with their 95% confidence intervals (CIs), were calculated for gait speed and the number of falls [11, 12]. The statistical heterogeneity was evaluated using the Cochran Q test and Higgins $I^2$ statistic. The random effects model was used when $I^2$ was $\geq$ 50%, and the fixed effects model was used when it was < 50%. The random-effects model created using the restricted maximum-likelihood estimator was used to obtain the pooled overall SMDs and 95% CIs for the outcomes [13].

Each moderator was subjected to meta-regression analysis for continuous variables (e.g., total number of patients, age, and proportion of female rate) and meta-analysis of variance for categorical variables (e.g., country) [12]. A restricted maximum-likelihood estimator was used to estimate the variance of the true effects to analyze potential moderators.

A two-sided P value of ≤ 0.05 or the absence of a null value (SMD = 0) within the 95% CIs were considered statistically significant. All analyses were conducted using R software version 4.3.1.

## Assessment of potential publication bias

Publication bias was analyzed using a funnel plot. The funnel plot is a schematic diagram of the SMD and standard error of the rivastigmine treatment. If there was no publication bias, individual studies were symmetrically distributed at the top of the funnel, whereas if there was a publication bias, they were relatively distributed outside the funnel if they showed asymmetry. In addition, summary statistics of publication bias were tested using the Egger linear regression test and Begg and Mazumdar rank correlation tests [12, 14, 15].

## Quality assessment

The risk of bias 2.0 (RoB) was developed in Cochrane Collaboration as a tool used to evaluate the quality of randomized controlled trial (RCT) studies [16]. A RoB consisted of high, low, or some concern for each evaluation item. The overall RoB evaluation of the individual studies is as follows: if all evaluation items are low, the overall evaluation is "low"; if there is only one concern of the evaluation items, the overall evaluation is "some concerns."; and if there are more than one concerns or even one high concern, the overall evaluation is "high concerns."

The Newcastle—Ottawa quality scale (NOS) was used to evaluate the quality of the case-control and cohort studies [17]. We assessed the following three parameters: (1) appropriate selection, (2) comparability of the research design or statistical analysis, and (3) outcome/exposure ascertainment and research procedures. We graded each parameter as a star; a study can be awarded a maximum of one star for each item in selection and outcome/exposure, but a maximum of two stars can be awarded for comparability. The quality of the evidence related to the estimation of benefits and disadvantages was displayed according to specific conditions.

# Results

## Study selection

A total of 222 articles were identified during the initial search across different electronic databases, 50 including PubMed (n = 23), Cochrane (n = 19), Embase (n = 139), Scopus (n = 38), and a manual search (n = 3). Of these, 65 studies were excluded because they either contained overlapping data or appeared in multiple databases. After reviewing the titles and abstracts, 130 studies were eliminated because they were unrelated or had trial registrations. Among the remaining 27 full-text articles, 17 were excluded for the following reasons: non-RCTs (n = 17). Ultimately, four studies, excluding three with mismatched outcomes and three without controls, met the selection criteria for qualitative and quantitative syntheses [5, 18–20] (S3 Table). Of these, three studies [5, 18, 20] contributed data to calculate the effect size for gait speed in rivastigmine treatment without a comparison group, and two studies [5, 19] provided data for fall numbers with a comparison group (Fig 1).

We performed a systematic review and meta-analysis of four studies, which included 286 participants (treatment group: 187 vs. control group: 99). A detailed description of the

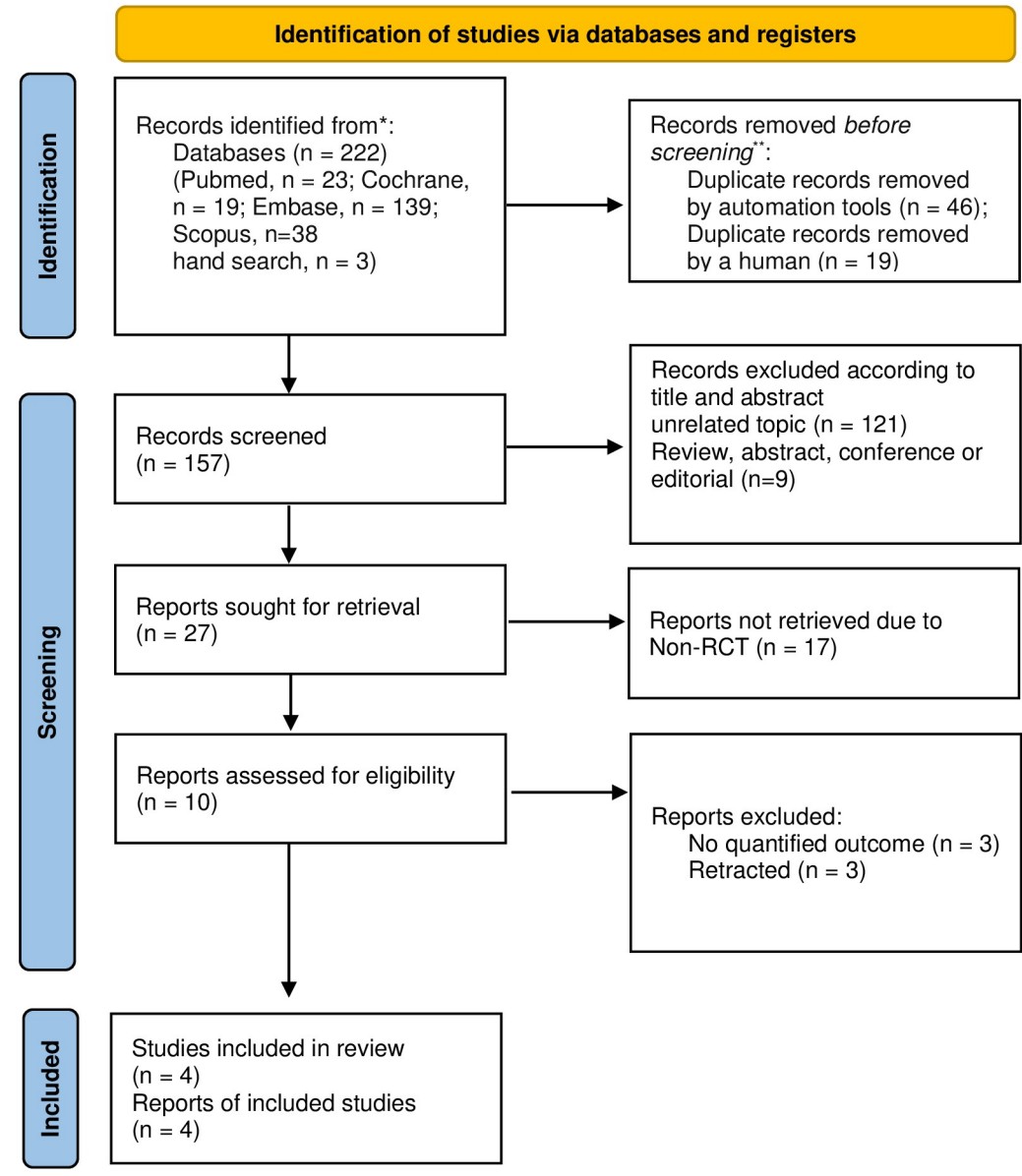

**Fig 1. PRISMA study selection flow chart.**

differences and participant characteristics is presented in Table 1. Two studies were randomized controlled trials (RCTs), while the other two were observational studies. The mean age of the participants ranged from 66.4 to 79.2 years, and the proportion of female participant rate ranged from 38.0 to 73.3%. Rivastigmine treatments administered included oral and patch formulations.

## Findings from meta-analysis

The pooled SMD for the overall gait speed from baseline without a comparison group was 1.260 (95% CI: −1.123 to 3.643), indicating no significant improvement in gait speed. The heterogeneity test resulted in a P value < 0.01 for Cochrane Q statistics, and Higgins' $I^2$ was

**Table 1. Characteristics of the included studies.**

| Study | Age (mean, year) | No.of patients | Female rate | Country | Outcome (unit) for gait | Study design | Follow-up (weeks) | Treatment/Intervention | Inclusion Criteria |
|---|---|---|---|---|---|---|---|---|---|
| Gurevich 2014 [18] | 79.2±5.9 (79.2) | 15 | 0.73 | Israel | TUG test(s) Gait speed (m/sec) Stride-time variability(%) | Prospective observational study | 12 | Rivastigmine oral, initially 1.5mg twice daily for 4weeks, 3mg twice daily for 4weeks, escalation 4.5mg twice daily until 12weeks | Non-demented elderly patient with HLGD |
| Shimura 2021 [20] | 79.03 ±6.89 (79.0) | 21 | 0.62 | Japan | Gait speed (m/min) Stride length (cm) Cadence (steps/min) | Prospective observational study | 12 | Rivastigmine patch, 9mg/day for 4weeks & 18mg/day for the subsequent 8weeks | Newly diagnosed mild to moderated AD>65years |
| Henderson 2016 [5] | 54–90* (70.0) | 65(65)** | 0.38 | UK | Gait speed (m/sec), Stride-time variability (sec) Fall number (per month) PPA fall risk score ICON-FES (fear of falling) Controlled leaning balance | RCT | 32 | Rivastigmine oral, 3mg per day (1.5mg twice daily) to the target dose of 12mg per day per day over 12weeks | PD without dementia, H&Y 2–3 |
| Li 2015 [19] | 66.8 ±12.2 (66.4) | 41(40)** | 0.38 | China | Fall number (per year) Incidence of Falls (%) | RCT | 48 | Rivastigmine oral, 3mg twice daily for 12month | PD with no CI, PD with CI (PD MCI + PDD) |

Abbreviations: AD, alzheimer's disease; CI, cognitive impairment; HLGD: higher-level gait disorder; H&Y, Hoehn and Yahr scale; ICON-FES, Iconographical Falls Efficacy Scale; MCI, mild cognitive impairment; No, number; PD, Parkinson's disease; PDD, Parkinson's disease dementia; PPA, Physiological Profile Assessment; RCT, Randomized controlled trials; TUG,Timed Up and Go; UK, United Kingdom.

*Range of age (Age is specified as a range in this study),

**Number of control group (equivalent as placebo)

97.4%. The pooled SMD for the overall number of falls between the rivastigmine treatment and control groups was −0.366 (95% CI: −0.650 to −0.083). The heterogeneity test resulted in a P value of 0.45 for Cochrane Q statistics, and Higgins' $I^2$ was 0%. Compared to the control group, the rivastigmine treatment group exhibited a significant reduction in the number of falls (Fig 2, S4 Table).

## Moderator analyses

This study explored the potential moderating roles of specific variables through meta-regression and meta-analysis of variance models. Table 2 presents the results of these analyses. Gait speed was significantly affected by age ($P < 0.001$) and female sex ($P = 0.004$). As age and proportion of female participants increased, gait speed decreased significantly. The number of patients ($P < 0.001$) and country ($P < 0.001$) also showed statistically significant impact on gait speed (Table 2).

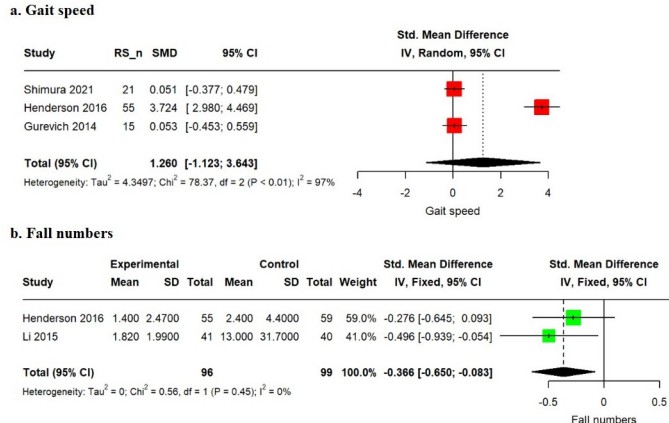

**Fig 2. Forest plots.** SMD, standardized mean difference. The random-effects model created using the restricted maximum-likelihood estimator. CI, confidence interval. The black diamond shows the overall effect size. In gait speed, the effect of the rivastigmine single group (pre-post). In fall numbers, the effect of the rivastigmine versus placebo.

## Publication bias

The statistical methods employed to detect publication bias or small-study effects are illustrated in S2 Fig. The individual SMD of gait speed showed visual asymmetry in the funnel plot. The p values for the Begg and Mazumdar rank correlation test (P = 0.117) and Egger linear regression coefficient test (P = 0.206) indicated no evidence of publication bias or small-study effects in this meta-analysis. As for the number of falls, the limited number of studies prevented confirmation of publication bias.

## Quality assessment

We evaluated the four included studies using a risk-of-bias 2.0 for the RCTs and NOS for the observational studies. With a risk of bias of 2.0, all studies were rated as having a "low" risk of bias. Li (2015) and Henderson (2016) were recognized as high-quality studies with low bias due to their randomization and blinding methods. Additionally, these studies had obtained prior IRB approval, resulting in minimal risk of missing data and reporting bias. In NOS, all studies were rated with a "Fair" grade. However, Gurevich (2014) and Shimura (2021) were

**Table 2. Effect of moderators for gait speed.**

| Variables | k | β | SMD | 95% CI | | P |
|---|---|---|---|---|---|---|
| | | | | **Gait speed** | | |
| No. of total patients | 3 | 0.097 | | 0.066 | 0.128 | <0.001 |
| Age | 3 | -0.403 | | -0.493 | -0.314 | <0.001 |
| Female rate | 3 | -11.101 | | -18.555 | -3.646 | 0.004 |
| Country | | | | | | <0.001 |
| Japan | 1 | | 0.051 | -0.377 | 0.479 | |
| Israel | 1 | | 0.053 | -0.453 | 0.559 | |
| UK | 1 | | 3.724 | 2.980 | 4.469 | |

Abbreviations: k, number of effect sizes; β, regression coefficient; SMD, standardized mean difference; P value from meta-regression analysis using the restricted maximum likelihood.

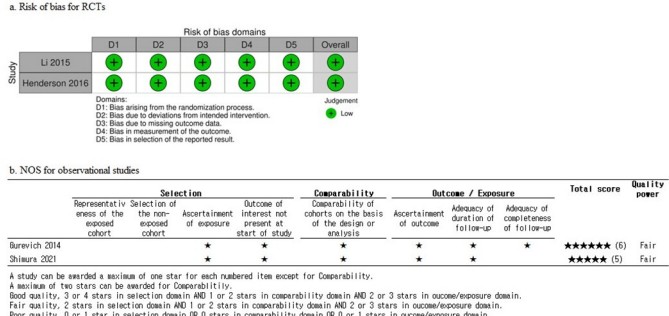

**Fig 3. Quality assessment.** Risk of bias for randomized controlled trials and Newcastle—Ottawa quality scale for observational studies.

deemed to have higher risk due to inadequate description of cohort representativeness in the exposed and non-exposed groups. Nevertheless, the overall assessment was considered "Fair" (Fig 3, S5 Table).

## Discussion

We conducted a literature review of published studies to confirm the effects of rivastigmine on gait of patients with neurodegenerative disorders. Subsequently, a meta-analysis was performed. Patient diagnoses in all studies, including two RCTs and two prospective observational studies, were established by neurologists (Table 1). The results of these four studies showed that treatment with rivastigmine in patients with neurodegenerative disorders had positive but insignificant effects on the improvement of gait speed and caused a significant decrease in fall occurrences.

Gait is an intricate process characterized by the automatic movement of legs controlled by the brainstem's locomotor generator, as well as the maintenance of balance, direction, and speed through the central and peripheral nervous systems [21]. In particular, in humans, the maintenance of cognitive control, especially sustained attention and executive function, plays a crucial role for gait [22]. Additionally, several neurotransmitter systems are involved in this process, but the cholinergic system in particular assumes a significant responsibility [23].

The major central cholinergic neurons consist of four subgroups, CH1-4, located in the basal forebrain, and two subgroups, CH5-6, located in the brainstem [24]. Among these, the forebrain cholinergic neurons serve as the major cholinergic inputs that project to the cerebral cortex, thalamus, and striatum, which are responsible for cognitive functions related to gait. Furthermore, it is well-established through numerous studies that such optimal cognitive input plays an inevitable role in maintaining stable gait and preventing falls [7, 25]. In contrast, cholinergic neurons in the pedunculopontine nucleus and laterodorsal tegmental nucleus extend projections to the thalamus, cerebellum, brainstem, and spinal cord, playing a crucial role in maintaining posture, balance, and gait [26]. Additionally, a decline in cholinergic levels with age has been reported to affect balance and gait disorders [7, 27]. Therefore, increasing acetylcholine levels via modulation of the cholinergic system is expected to yield favorable outcomes in enhancing gait and reducing fall risk, especially among older individuals and those with neurodegenerative disease marked by cholinergic degeneration [23]. This perspective aligns with observations indicating that dopaminergic replacement treatment has no impact on gait and postural control in patients with PD, prompting a shift in research focus towards

investigating the potential benefits of AChE inhibitors linked to cholinergic regulation [9, 28, 29]. For this reason, there is considerable ongoing research aimed at investigating basic experimental models related to the modulation of the cholinergic system and identifying therapeutic targets in degenerative brain diseases such as AD and PD [30–32].

In the human brain, cholinergic regulation is controlled by two cholinesterases (ChEs), AChE and BChE [33]. AChE is the primary cholinesterase found predominantly at nerve synaptic junctions and exhibits robust activity in the adult human cerebral cortex. Conversely, BChE is primarily localized in glial cells. In patients with AD, the gradual loss of neurons in the cerebral cortex leads to decreased AChE activity and increased BChE activity. This indicates a significant role of BChE in the regulation of brain acetylcholine levels [34]. This finding has also been reported in several preclinical studies, where preclinical trials targeting AChE-knockout mice revealed that BChE functions in the brain and plays a predominant role in the hydrolysis of acetylcholine [35].

Moreover, according to studies using ChE-knockout mice, injection of rivastigmine increased hippocampal acetylcholine levels, whereas donepezil did not. This suggests that rivastigmine inhibits BChE, leading to an increase in acetylcholine levels [36].

Drugs currently used for cholinergic regulation in clinical practice include donepezil, galantamine, and rivastigmine. While donepezil and galantamine selectively inhibit AChE, rivastigmine acts as a dual inhibitor, targeting both AChE and BuChE. Previous preclinical and clinical research findings indicated that modulating both AChE and BuChE through dual inhibition may offer superior treatment efficacy compared to selectively targeting AChE alone [35].

In our study, although the use of rivastigmine did not yield a significant difference, gait speed tended to improve (Fig 2). Furthermore, the moderator analysis revealed that gait speed tended to decrease with increasing age, and the decrease was slower in females (Table 2). This observation aligns with the widely acknowledged findings from various studies related to gait speed [37, 38]. Gait speed is widely recognized as a key indicator in gait analysis, with a strong association with acetylcholine levels [22]. This suggests that gait involves higher cognitive functions, with executive function and attention being particularly crucial for maintaining safety and executing deliberate movements during walking [2]. Although our study did not reveal a significant increase in gait speed, this discrepancy may be attributed to differences in gait measurement methods across studies and variations in the conditions of the included study populations. Therefore, with future well-designed RCTs, factors influencing gait speed can be evaluated more accurately.

Furthermore, the meta-analysis results on falls showed a SMD of −0.366 (95% CI: −0.650 to −0.083), indicating a significant effect of rivastigmine on falls, with a reduction of approximately 20.4% in the number of falls among patients receiving rivastigmine (Fig 2). This result is consistent with those of previous studies indicating that reduced acetylcholine activity is associated with fall incidence and that the use of donepezil or rivastigmine significantly reduces fall rates [39]. Additionally, Lord et al. reported that in patients with PD, an increase in falls is associated with poor performance in attention tasks, suggesting that while the brainstem cholinergic system is a primary neural correlate of falls, cognition associated with neocortical acetylcholine originating from the basal forebrain, particularly attention, may play a significant role in fall occurrence [40]. The precise mechanism of the cognitive contribution to gait and falls remains unclear; however, through previous evidence and future studies, a more detailed mechanism will be elucidated.

This study has some limitations. First, heterogeneity was inevitable among included patients. The study included patients diagnosed with neurodegenerative disorders accompanied by abnormalities in the cholinergic system, such as AD, PD, and higher level gait

disorders; however, it could not account for the differences in the pathogenesis of each condition. Second, the absence of individual patient data stratified by varying degrees of cognitive decline and motor severity is a limitation of this study. Additionally, owing to the differences in gait parameters, controls, and methods of gait assessment employed in each study, our analysis was restricted to a simple gait analysis without control group. This limitation arises from the scarcity of RCTs investigating the association between rivastigmine use and gait. Therefore, to elucidate the effects of rivastigmine on gait and falls, well-designed large-scale RCT trials are needed. Moreover, beyond the currently used donepezil, galantamine, and rivastigmine, research is anticipated on pharmacotherapeutic targets that possess AChE-inhibiting properties and could improve gait and reduce falls in patients with neurodegenerative diseases, thereby enhancing cholinergic function [41, 42].

Despite these limitations, our study represents the first meta-analysis to investigate the impact of dual-ChEIs on gait and falls in patients with neurodegenerative diseases. Numerous studies have investigated the effects of rivastigmine on patients with PD and AD. However, most of these studies have focused on cognition. Furthermore, while there have been meta-analytic studies on the effects of ChEIs on gait and falls, research specifically aimed at confirming the effects of rivastigmine, a dual ChEI, on gait and falls has not yet been conducted. Although the studies included in the analysis did not have identical study methods or complete information, through the advantages or assistance of meta-analysis, we confirmed the benefits of rivastigmine in improving gait speed and reducing falls.

## Conclusion

Treatment with dual-ChEIs, particularly rivastigmine, tends to improve gait speed and significantly reduce the incidence of falls in patients with neurodegenerative disorders. Considering the limited efficacy of current interventions for gait disturbances and falls, these findings highlight the potential of dual-ChEIs as a promising treatment option. However, limitations include patient heterogeneity, differences in gait measurement methods, and a lack of randomized controlled trials (RCTs) specifically investigating rivastigmine's effects on gait. Future well-designed large-scale RCTs are essential to comprehensively elucidate these effects and to clarify the precise associations and mechanisms in patients with neurodegenerative disorders characterized by high fall risk and gait disturbances.

## Supporting information

**S1 Table. PRISMA checklist.**
(PDF)

**S2 Table. Search queries.**
(PDF)

**S3 Table. Included and excluded studies in full-text analysis.**
(PDF)

**S4 Table. Characteristics of the included studies.**
(PDF)

**S5 Table. The completed risk of bias and quality/certainty assessments for each study.**
(PDF)

**S1 Fig. Flow chart of a quantitative meta-analysis.**
(JPG)

**S2 Fig. Funnel plot.**
(PDF)

## Author Contributions

**Conceptualization:** Sung Ryul Shim, Jong-Yeup Kim, Seon-Min Lee.

**Data curation:** Sung Ryul Shim, Yungjin Lee, Seon-Min Lee.

**Formal analysis:** Sung Ryul Shim, Jong-Yeup Kim, Yungjin Lee, Seon-Min Lee.

**Funding acquisition:** Seon-Min Lee.

**Investigation:** Sung Ryul Shim, Jong-Yeup Kim, Seon-Min Lee.

**Methodology:** Sung Ryul Shim, Seon-Min Lee.

**Project administration:** Jong-Yeup Kim, Jieun Shin, Seon-Min Lee.

**Software:** Sung Ryul Shim, Seon-Min Lee.

**Supervision:** Kyum-Yil Kwon.

**Validation:** Sung Ryul Shim, Jong-Yeup Kim, Seon-Min Lee.

**Visualization:** Sung Ryul Shim, Jong-Yeup Kim, Jieun Shin, Seon-Min Lee.

**Writing – original draft:** Seon-Min Lee.

**Writing – review & editing:** Seon-Min Lee.

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
