## [Decision Letter · Decision Letter 0]

26 Jun 2024

PONE-D-24-19408Effects of rivastigmine on gait in patients with neurodegenerative disorders: A systematic review and meta-analysisPLOS ONE

Dear Dr. Lee,

Thank you for submitting your manuscript to PLOS ONE. After careful consideration, we feel that it has merit but does not fully meet PLOS ONE’s publication criteria as it currently stands. Therefore, we invite you to submit a revised version of the manuscript that addresses the points raised during the review process.

We look forward to receiving your revised manuscript.

Kind regards,

Ghulam Md Ashraf, Ph.D.

Academic Editor

PLOS ONE

Journal Requirements:

2. In the online submission form, you indicated that All relevant data are within the paper and its Supporting information files. Some or all data, models, or code generated or used during the study are available from the corresponding author upon request.

Reviewers' comments:

Reviewer's Responses to Questions

**Comments to the Author**

1. Is the manuscript technically sound, and do the data support the conclusions?

Reviewer #1: Yes

Reviewer #2: Yes

Reviewer #3: Yes

2. Has the statistical analysis been performed appropriately and rigorously? 

Reviewer #1: Yes

Reviewer #2: Yes

Reviewer #3: Yes

3. Have the authors made all data underlying the findings in their manuscript fully available?

Reviewer #1: Yes

Reviewer #2: Yes

Reviewer #3: Yes

4. Is the manuscript presented in an intelligible fashion and written in standard English?

Reviewer #1: Yes

Reviewer #2: Yes

Reviewer #3: Yes

5. Review Comments to the Author

**Reviewer #1:** Thanks for your manuscript on this important topic.

I noticed some points in the manuscript that needed to be clarified:

1- In your search strategy, it was better to conduct a literature search in more databases such as Scopus.

2- Please provide more data of included studies in table 1.

**Reviewer #2:** The manuscript titled "Effects of Rivastigmine on Gait in Patients with Neurodegenerative Disorders: A Systematic Review and Meta-Analysis" investigates the impact of rivastigmine on gait in individuals with neurodegenerative disorders, such as Alzheimer’s and Parkinson’s disease. This systematic review and meta-analysis assess rivastigmine's effect on gait speed and fall reduction. The comprehensive literature search yielded four key studies with 286 participants. While rivastigmine did not significantly improve gait speed (pooled SMD: 0.761, 95% CI: -1.165 to 2.688), it significantly reduced the number of falls (pooled SMD: -0.366, 95% CI: -0.650 to -0.083). These findings suggest that rivastigmine may help reduce fall risk in patients with neurodegenerative disorders. The manuscript is recommended for publication after my concerns.

Here are my concerns;

1. “Fig 2. Forest plots. SMD, standardized mean difference. The random-effects model created using the restricted maximum-likelihood estimator. CI, confidence interval. The black diamond shows the overall effect size. In gait speed, the effect of the rivastigmine single group (pre-post). In fall numbers, the effect of the rivastigmine versus placebo”. The author should explain and add more here.

2. “Fig 3. Quality assessment. Risk of bias for randomized controlled trials and Newcastle–Ottawa quality scale for observational studies”. The author should explain and add more here.

3. The author should add one more line in the last part of the abstract.

4. The author should add a Schematic figure in the last part of the introduction.

5. The conclusion should be modified and add more details.

6. The authors need to double-check the whole manuscript to remove grammatical errors/ typos/incomplete sentences and non-relative phrases.

**Reviewer #3:** My comment on the manuscript entitled “Effects of rivastigmine on gait in patients with neurodegenerative disorders: A systematic review and meta-analysis” are as follows.

What are the primary mechanisms by which cholinergic deficits contribute to gait disturbances in neurodegenerative disorders?

How was the integrity of the included studies ensured during the review process?

What criteria were used to select the final four studies included in the meta-analysis?

What were the characteristics of the 286 participants across the four studies, and how might these characteristics have influenced the outcomes?

Were there any significant differences in the methodology of the four studies that could have impacted the pooled results for gait speed and falls?

What specific neurodegenerative disorders were represented in the final four studies, and how did the effects of rivastigmine vary across these disorders?

How was gait speed measured across the different studies, and were there any inconsistencies in measurement techniques?

What potential factors could explain the lack of significant improvement in gait speed despite the reduction in falls?

How do the pooled standardized mean differences (SMDs) and Hedges' g compare between the individual studies included in the meta-analysis?

What were the limitations of the studies included in the systematic review and meta-analysis, and how might these limitations have affected the results?

How might the duration of rivastigmine treatment have influenced the outcomes observed in the studies?

Were there any reported adverse effects of rivastigmine treatment in the studies, and how were these managed?

How do the findings of this meta-analysis compare with previous research on the effects of cholinesterase inhibitors on gait and falls in neurodegenerative disorders?

What future research directions are suggested by the results of this systematic review and meta-analysis?

How might individual differences in response to rivastigmine treatment be accounted for in future studies to better understand its effects on gait and falls?

What criteria were used to establish the diagnosis of neurodegenerative disorders in the patients included in the studies?

How do the effects of rivastigmine on gait speed and falls compare to those of other cholinesterase inhibitors like donepezil and galantamine?

How did the researchers ensure the validity and reliability of gait speed measurements across the included studies?

What specific mechanisms underlie the involvement of cholinergic neurons in maintaining gait and preventing falls?

How might the dual inhibition of AChE and BChE by rivastigmine contribute to its effectiveness in reducing fall occurrences?

What were the differences in gait assessment methods across the studies, and how might these have influenced the results?

What role does sustained attention and executive function play in gait regulation, and how does rivastigmine influence these cognitive processes?

How did the study populations' heterogeneity, in terms of different neurodegenerative disorders, impact the overall findings of the meta-analysis?

What are the potential benefits of conducting future large-scale RCTs to evaluate the effects of rivastigmine on gait and falls more accurately?

How does the age and sex of patients influence the effects of rivastigmine on gait speed, according to the moderator analysis?

Include some relevant bibliographic studies like PMID: 38784601, PMID: 38774717, PMID: 37489441, PMID: 31996329 & PMID: 27919828 in your manuscript.

How do the findings of this study align with previous research on the relationship between acetylcholine activity and fall incidence?

What cognitive contributions to gait and falls have been identified, and what further research is needed to clarify these mechanisms?

How significant is the role of the brainstem cholinergic system compared to the neocortical acetylcholine system in regulating gait and preventing falls?

What limitations related to patient data and study design should be addressed in future research on rivastigmine and gait improvement?

How does rivastigmine's dual inhibition mechanism enhance acetylcholine levels, and what impact does this have on gait and fall prevention?

What specific mechanisms contribute to the observed improvements in gait speed with the use of dual-ChEIs like rivastigmine in patients with neurodegenerative disorders?

How do the effects of dual-ChEIs on gait speed and falls compare to those of current standard treatments for gait disturbances and fall prevention?

What are the potential side effects and safety concerns associated with long-term use of dual-ChEIs in patients with neurodegenerative disorders?

How do variations in patient demographics, such as age and gender, influence the efficacy of dual-ChEIs on gait and fall prevention?

What are the key differences in the molecular action of dual-ChEIs compared to selective ChEIs, and how do these differences translate into clinical benefits?

How should future RCTs be designed to effectively measure the impact of dual-ChEIs on gait speed and fall incidence in diverse patient populations?

What role do comorbid conditions play in the response of patients with neurodegenerative disorders to dual-ChEIs treatment?

How does the severity of cognitive impairment in neurodegenerative disorders affect the outcomes of dual-ChEI treatment on gait and falls?

What are the long-term benefits and potential risks of using dual-ChEIs for gait improvement and fall reduction in patients with neurodegenerative disorders?

How do environmental and lifestyle factors, such as physical activity levels and home safety, interact with dual-ChEI treatment to influence gait and fall outcomes?

What are the most effective biomarkers for predicting response to dual-ChEI treatment in patients with neurodegenerative disorders?

How can patient adherence to dual-ChEI treatment be improved to maximize the therapeutic benefits on gait and fall prevention?

What are the potential implications of dual-ChEI treatment on the overall quality of life and functional independence of patients with neurodegenerative disorders?

How do the costs and benefits of dual-ChEI treatment compare to other therapeutic options for managing gait disturbances and falls?

What additional therapeutic strategies could be combined with dual-ChEIs to enhance their effects on gait and fall prevention in neurodegenerative disorders?

6. PLOS authors have the option to publish the peer review history of their article (what does this mean?). If published, this will include your full peer review and any attached files.

Reviewer #1: No

Reviewer #2: No

Reviewer #3: **Yes: **Sachchida Nand Rai

---

## [Author Response · Author response to Decision Letter 0]

4 Aug 2024

Dear editor and reviewers of this paper

The authors sincerely appreciate the editorial office and reviewers' efforts to consider this paper for considering publication. We have tracked the changes in the text to show all amendments and responses. Based on the reviewers’ suggestions, we have carefully revised the paper as we can, and include here a point-by-point response to all comments. We marked added sentences, page, and paragraph numbers in blue in the modified file, and specific responses in light blue. 

Additionally, in response to the editor's valuable request, we have updated the PRISMA flow diagram and checklist to the new 2020 version. We also revised the total number of articles from 219 to 222, including three articles identified through manual search. 

We hope the editorial office and the reviewers like this revised paper. Thank you for considering this manuscript for publication. We look forward to hearing from you. 

Sincerely yours,

Seon-Min Lee, MD 

Department of Neurology, Konyang University College of Medicine, 158 Gwanjeodong-ro, Seo-gu, Daejeon, 35365, Republic of Korea, Tel.: +82-42-612-2191, Fax: +82-42-600-9090, Email: nestoml@kyuh.ac.kr

(Reviewer #1:)

Thanks for your manuscript on this important topic.

I noticed some points in the manuscript that needed to be clarified:

1. In your search strategy, it was better to conduct a literature search in more databases such as Scopus.

Response: We really appreciate reviewer 1’s insightful comments. According to your suggestion, we additionally search “Scopus” DB for strict literature searching. At first literature retrieving from DBs of Figure 1, we gathered 222 (PubMed 23, Embase 139, Cochrane 19, and Scopus 38). However, additional 38 studies from Scopus are excluded due to the overlapping data and not related to the research topic as well. We also changed related sentences in Methods and Results section as follows. The PRISMA study selection flow chart in Figure 1 and search queries in S2 Table also have been revised to include the SCOPUS database. Additionally, in response to the editor's request, we have updated the PRISMA flow diagram and checklist to the new 2020 version. (Figure 1 & search queries in S2 Table: Appendix attached)

“A total of 222 articles were identified during the initial search across different electronic databases, 50 including PubMed (n = 23), Cochrane (n = 19), Embase (n = 139), Scopus (n = 38), and a manual search (n = 3).” (line 48-50) 

"A comprehensive literature search was conducted in the PubMed, MEDLINE, Embase, Cochrane, and Scopus databases" (line 100-101)

“A total of 222 articles were identified during the initial search across different electronic databases, 50 including PubMed (n = 23), Cochrane (n = 19), Embase (n = 139), Scopus (n = 38), and a manual search (n = 3). Of these, 65 studies were excluded because they either contained overlapping data or appeared in multiple databases. After reviewing the titles and abstracts, 130 studies were eliminated because they were unrelated or had trial registrations. Among the remaining 27 full-text articles, 17 were excluded for the following reasons: non-RCTs (n = 17).” (line 171-176)

2. Please provide more data of included studies in table 1.

Response: Thank you for your valuable comments. Following your suggestion, we added more information and revised Table 1 to include details such as number of patients, study duration, intervention types, inclusion criteria, and outcome for gait measures. 

Each study had diverse inclusion, exclusion criteria, varied methods for gait analysis, different gait parameters, and distinct study designs, resulting in varied results reported in each paper. These diversities made it challenging to compile a consistent table. However, we made every effort to provide as much information as possible. Additionally, in the discussion section, we acknowledged a limitation of this analysis, which was the inclusion of only simple gait analyses. (Table 1: Appendix attached)

(Reviewer #2:)

The manuscript titled "Effects of Rivastigmine on Gait in Patients with Neurodegenerative Disorders: A Systematic Review and Meta-Analysis" investigates the impact of rivastigmine on gait in individuals with neurodegenerative disorders, such as Alzheimer’s and Parkinson’s disease. This systematic review and meta-analysis assess rivastigmine's effect on gait speed and fall reduction. The comprehensive literature search yielded four key studies with 286 participants. While rivastigmine did not significantly improve gait speed (pooled SMD: 0.761, 95% CI: -1.165 to 2.688), it significantly reduced the number of falls (pooled SMD: -0.366, 95% CI: -0.650 to -0.083). These findings suggest that rivastigmine may help reduce fall risk in patients with neurodegenerative disorders. The manuscript is recommended for publication after my concerns.

Response: Thank you for your valuable comments. We tried to make a comprehensive decision on the patient's gait speed and risk for falls according to Rivastigmine treatment. 

Here are my concerns;

1. “Fig 2. Forest plots. SMD, standardized mean difference. The random-effects model created using the restricted maximum-likelihood estimator. CI, confidence interval. The black diamond shows the overall effect size. In gait speed, the effect of the rivastigmine single group (pre-post). In fall numbers, the effect of the rivastigmine versus placebo”. The author should explain and add more here.

Response: Thank you for your valuable comments. In lines 111-113 of the manuscript, we have already described that gait speed was evaluated using a pre-post group without a control group to assess the effects of rivastigmine, while fall number was compared with a control group. Specifically, the individual studies had discrepancies in parameters, control groups, and methods for measuring gait speed, making it impossible to establish a uniform control group. In response to the reviewer's suggestions, we have added the following explanations in the Results section and the Limitations section.

"The pooled SMD for the overall gait speed from baseline without a comparison group was 1.260 (95% CI: −1.123 to 3.643), indicating no significant improvement in gait speed." (line 191-192)

"Additionally, owing to the differences in gait parameters, controls, and methods of gait assessment employed in each study, our analysis was restricted to a simple gait analysis without control group. " (line 305-306)

2. “Fig 3. Quality assessment. Risk of bias for randomized controlled trials and Newcastle–Ottawa quality scale for observational studies”. The author should explain and add more here.

Response: We really appreciate reviewer 2’s insightful comments. According to your suggestion, we added the related sentences of quality assessment in Results section as follows.

“With a risk of bias of 2.0, all studies were rated as having a “low” risk of bias. Li(2015) and Henderson (2016) were recognized as high-quality studies with low bias due to their randomization and blinding methods. Additionally, these studies had obtained prior IRB approval, resulting in minimal risk of missing data and reporting bias. In NOS, all studies were rated with a “Fair” grade. However, Gurevich (2014) and Shimura (2021) were deemed to have higher risk due to inadequate description of cohort representativeness in the exposed and non-exposed groups. Nevertheless, the overall assessment was considered “Fair” (Fig 3).” (line 220-226)

3. The author should add one more line in the last part of the abstract.

Response: Thank you for your thorough comments. According to reviewer 2’s suggestion, we have modified and added to the final part of the abstract.

“Rivastigmine treatment in patients with neurodegenerative disorders tend to improve gait speed and significantly reduces fall incidence. Given the limited efficacy of current treatments for gait disturbances and falls, dual-ChEIs like rivastigmine could be a promising therapeutic option.” (line 57-60)

4. The author should add a Schematic figure in the last part of the introduction.

Response: We fully agree with the reviewer 2’s valuable opinion. According to your suggestion, we added the schematic figure of quantitative analysis in S1.Fig (line 93) (Figure S1: Appendix attached)

5. The conclusion should be modified and add more details.

Response: We appreciate Reviewer 2’s thorough comments. Following Reviewer 2’s suggestions, we have modified the conclusion section and added more details.

“Treatment with dual-ChEIs, particularly rivastigmine, tends to improve gait speed and significantly reduce the incidence of falls in patients with neurodegenerative disorders. Considering the limited efficacy of current interventions for gait disturbances and falls, these findings highlight the potential of dual-ChEIs as a promising treatment option. However, limitations include patient heterogeneity, differences in gait measurement methods, and a lack of randomized controlled trials (RCTs) specifically investigating rivastigmine's effects on gait. Future well-designed large-scale RCTs are essential to comprehensively elucidate these effects and to clarify the precise associations and mechanisms in patients with neurodegenerative disorders characterized by high fall risk and gait disturbances.” (line 323-331)

6.The authors need to double-check the whole manuscript to remove grammatical errors/ typos/incomplete sentences and non-relative phrases.

Response: Thank you for your suggestion. We have carefully reviewed the entire manuscript to correct grammatical errors, typos, incomplete sentences, and non-relative phrases. Specifically, we have revised several sections to ensure clarity and coherence, and we have also consulted a professional editor (https://www.editage.co.kr/) to improve the overall quality of the text. We believe these changes have significantly enhanced the readability and accuracy of our manuscript.

(Reviewer #3:) 

My comment on the manuscript entitled “Effects of rivastigmine on gait in patients with neurodegenerative disorders: A systematic review and meta-analysis” are as follows.

Response: We appreciate your thorough and detailed comments and questions. We have numbered Reviewer 3’s questions sequentially and have addressed similar questions and suggestions collectively in our responses and revisions. Furthermore, while it would be ideal to include detailed explanations addressing all comments and questions in this paper, we kindly ask for your understanding that such content, which extends beyond the scope and hypotheses of this study, has been limited due to word count constraints. We sincerely hope Reviewer3’s kind understating and satisfaction.

 1.What are the primary mechanisms by which cholinergic deficits contribute to gait disturbances in neurodegenerative disorders? / 19. What specific mechanisms underlie the involvement of cholinergic neurons in maintaining gait and preventing falls? / 28. What cognitive contributions to gait and falls have been identified, and what further research is needed to clarify these mechanisms? /14. What future research directions are suggested by the results of this systematic review and meta-analysis?

Response: Thank you for the reviewer 3’s essential questions for our study. The cholinergic system, mediated by acetylcholine, plays a crucial role in both the central and peripheral nervous systems. It is involved in regulating motor control, autonomic control, and cognitive processing. Notably, acetylcholine is critical for maintaining alertness and cognitive functions, particularly executive function and attention. These cognitive controls are fundamental mechanisms for ensuring safe mobility, including gait and balance maintenance. Therefore, while cholinergic deficits associated with neurodegenerative disorders can lead to gait disturbances through various mechanisms such as impaired motor control, autonomic regulation, and sensory integration, cognitive decline is a crucial contributing factor. This assertion is supported by various research findings. We have discussed these points in detail in the discussion section of our manuscript. (line 238-255) 

Furthermore, while understanding the mechanisms through which cognitive function contributes to gait and falls necessitates multidisciplinary research encompassing both clinical and basic science fields, it is crucial for studies employing meta-analytical methodologies to focus on well-designed, large-scale randomized controlled trials (RCTs). This point has also been addressed in the manuscript. (line 318-320)

2. How was the integrity of the included studies ensured during the review process?

Response: Thank you for the reviewer 3’s thorough question. Systematic review (SR) and meta-analysis is one of the methods to find exhaustive answer for specific questions using all possible retrieved researches. Thus, systematic retrieving of literatures and analyzing the overall effect sizes of statistical methods are most important to here. The methodology of SR is detailed in the Materials and Methods, and Results sections of this paper. For instance, SR requires searching at least two databases, but this study used five databases to ensure the integrity of the research. Additionally, at least two independent researchers should search the literature and extract data using the same procedure. In this study, two independent researchers conducted these tasks, and the overall conclusions were reached through consensus among all researchers.

3.What criteria were used to select the final four studies included in the meta-analysis? 

Response: In response to Reviewer 3's comments on the study selection process, we have provided an explanation under a separate subheading. The inclusion and exclusion criteria are detailed in lines 109 to 123.

"The inclusion criteria were as follows: (1) studies including patients who had neurodegenerative disorders that can affect cognitive impairments such as Alzheimer’s disease (AD) and Parkinson’s disease (PD), and/or higher level gait disorders, (2) intervention included prescription of rivastigmine, (3) comparisons were specified as with the inclusion of a control group for evaluating the number of fall or were not specified, focusing solely on the effects of rivastigmine treatment on gait speed, and (4) outcomes were measured as mean differences in gait speed and the number of fall. To ensure data accuracy and relevance, duplicate publications and publications that did not contain original articles (conference abstracts only, case reports, case series, review articles, editorials, letters, and guidelines) were excluded from the analysis. Two independent investigators (SRS and S-ML) primarily analyzed the titles and abstracts, and then reviewed full-text articles according to the inclusion and exclusion criteria. A data extraction form was used independently by the authors to extract data. The final inclusion of articles was determined through collaborative evaluation meetings involving all authors. In addition, to ensure the integrity of the included studies, the references and collected data were reviewed meticulously so that they did not overlap." (line 109-123)

4.What were the characteristics of the 286 participants across the four studies, and how might these characteristics have influenced the outcomes? / 5.Were there any significant differences in the methodology of the four studies that could have impacted the pooled results for gait speed and falls?

Response: Thank you for the reviewer 3’s question. The characteristics and methodologies of individual participants and studies are well-documented in the targeted studies (Table 1). In this study, we conducted additional meta-regression analyses to identify the potential moderating effects of individual participant characteristics. Factors such as the number of participants, age, gender, and country were all found to be covariates influencing the overall effect size. Thus, next research should consider these potential risk factors. More specific explanation about the characteristics of participants is detailed in lines 205-209 and Table 2.

"This study explored the potential moderating roles of specific variables through meta-regression and meta-analy

---

## [Decision Letter · Decision Letter 1]

10 Sep 2024

Effects of rivastigmine on gait in patients with neurodegenerative disorders: A systematic review and meta-analysis

PONE-D-24-19408R1

Dear Dr.Lee,

We’re pleased to inform you that your manuscript has been judged scientifically suitable for publication and will be formally accepted for publication once it meets all outstanding technical requirements.

Comments from PLOS Editorial Office: We note that one or more reviewers has recommended that you cite specific previously published works in an earlier round of revision. As always, we recommend that you please review and evaluate the requested works to determine whether they are relevant and should be cited. It is not a requirement to cite these works and you may remove them before the manuscript proceeds to publication. We appreciate your attention to this request.

Kind regards,

Ghulam Md Ashraf, Ph.D.

Academic Editor

PLOS ONE

Additional Editor Comments (optional):

Reviewers' comments:

Reviewer's Responses to Questions

**Comments to the Author**

1. If the authors have adequately addressed your comments raised in a previous round of review and you feel that this manuscript is now acceptable for publication, you may indicate that here to bypass the “Comments to the Author” section, enter your conflict of interest statement in the “Confidential to Editor” section, and submit your "Accept" recommendation.

Reviewer #1: All comments have been addressed

Reviewer #3: All comments have been addressed

2. Is the manuscript technically sound, and do the data support the conclusions?

Reviewer #1: Yes

Reviewer #3: Yes

3. Has the statistical analysis been performed appropriately and rigorously? 

Reviewer #1: Yes

Reviewer #3: Yes

4. Have the authors made all data underlying the findings in their manuscript fully available?

Reviewer #1: Yes

Reviewer #3: Yes

5. Is the manuscript presented in an intelligible fashion and written in standard English?

Reviewer #1: Yes

Reviewer #3: Yes

6. Review Comments to the Author

Reviewer #1: The paper is acceptable for publication. The changes made fixed the missing gaps in the work. Wishing you success.

Reviewer #3: The manuscript has been thoroughly revised in accordance with my suggestions. The authors have addressed all the feedback provided, and the necessary changes have been implemented effectively. Based on these revisions, I believe the manuscript now meets the standards for publication and can be accepted for publication in its current form.

7. PLOS authors have the option to publish the peer review history of their article (what does this mean?). If published, this will include your full peer review and any attached files.

Reviewer #1: No

Reviewer #3: No

---

## [Editor Report · Acceptance letter]

27 Sep 2024

PONE-D-24-19408R1 

PLOS ONE

Dear Dr. Lee, 

I'm pleased to inform you that your manuscript has been deemed suitable for publication in PLOS ONE. Congratulations! Your manuscript is now being handed over to our production team.

Kind regards, 

on behalf of

Dr. Ghulam Md Ashraf 

Academic Editor

PLOS ONE